# Recent Advances in the Therapeutic Potential of Carotenoids in Preventing and Managing Metabolic Disorders

**DOI:** 10.3390/plants13121584

**Published:** 2024-06-07

**Authors:** Ana E. Ortega-Regules, Juan Alonso Martínez-Thomas, Karen Schürenkämper-Carrillo, Cecilia Anaya de Parrodi, Edgar R. López-Mena, Jorge L. Mejía-Méndez, J. Daniel Lozada-Ramírez

**Affiliations:** 1Departamento de Ciencias de la Salud, Universidad de las Américas Puebla, Ex Hacienda Sta. Catarina Mártir S/N, Puebla 72810, San Andrés Cholula, Mexico; ana.ortega@udlap.mx; 2Departamento de Ciencias Químico-Biológicas, Universidad de las Américas Puebla, Ex Hacienda Sta. Catarina Mártir S/N, Puebla 72810, San Andrés Cholula, Mexico; juan.martinezts@udlap.mx (J.A.M.-T.); karen.schurenkamperco@udlap.mx (K.S.-C.); cecilia.anaya@udlap.mx (C.A.d.P.); 3Tecnologico de Monterrey, Escuela de Ingeniería y Ciencias, Av. Gral. Ramón Corona No 2514, Zapopan 45121, Colonia Nuevo México, Mexico; edgarl@tec.mx

**Keywords:** carotenoids, metabolic disorders, encapsulation, biological activity, stability, bioavailability

## Abstract

Carotenoids constitute compounds of significant biological interest due to their multiple biological activities, such as antimicrobial, anticancer, antiadipogenic, antidiabetic, and antioxidant properties. Metabolic syndrome (MetS) comprehends a series of metabolic abnormalities (e.g., hypertension, obesity, and atherogenic dyslipidemia) that can affect children, adolescents, and the elderly. The treatment of MetS involves numerous medications, which, despite their efficacy, pose challenges due to prolonged use, high costs, and various side effects. Carotenoids and their derivatives have been proposed as alternative treatments to MetS because they reduce serum triglyceride concentrations, promote insulin response, inhibit adipogenesis, and downregulate angiotensin-converting enzyme activity. However, carotenoids are notably sensitive to pH, light exposure, and temperature. This review addresses the activity of carotenoids such as lycopene, lutein, fucoxanthin, astaxanthin, crocin, and *β*-carotene towards MetS. It includes a discussion of sources, extraction methods, and characterization techniques for analyzing carotenoids. Encapsulation approaches are critically reviewed as alternatives to prevent degradation and improve the biological performance of carotenoids. A brief overview of the physiopathology and epidemiology of the diseases, including MetS, is also provided.

## 1. Introduction

Metabolic syndrome (MetS), a complex of physiological and metabolic disorders, is a significant health concern. It includes atherogenic dyslipidemia, dysglycemia, abdominal obesity, cardiovascular diseases, hyperglycemia, and fatty liver [1]. The development of MetS is influenced by both physiological and behavioral factors. Smoking, poor diet, and limited physical activity fall under the former, while high blood cholesterol levels and high body mass index are associated with the latter [2].

The prevalence of MetS can vary depending on factors such as age, socioeconomic status, ethnic groups, gender, and diagnosed disease [3]. Based on the latter, it has been estimated that MetS affects approximately 20–25% of global adults and 0–19.2% of children [4,5]. Current treatment approaches for MetS are primarily focused on changes in dietary habits and the intensity of physical exercises [6]. Regarding drug therapy, commonly prescribed medications for the treatment of MetS include angiotensin-converting enzyme (ACE) inhibitors, angiotensin receptor blockers (ARBs), statins, and metformin [7].

Carotenoids represent a unique class of organic molecules extensively present in higher plants, algae, bacteria, and fungi. Depending on the organisms that synthesize them, carotenoids can regulate internal signaling mechanisms, mediate stress responses, and modulate complex biochemical processes such as photosynthesis [8]. In nature, carotenoids can be found in peaches, papaya, cereal grains, dairy goods, and insects [9]. Despite their wide distribution, carotenoids’ biosynthesis can be influenced by high or low altitude, climate conditions, humidity, and exposure to light [10,11].

From biological sources, carotenoids can be extracted through standard laboratory techniques such as maceration and Soxhlet extraction using non-polar (such as hexane and petroleum ether) and polar solvents (such as acetone and ethanol), separated using chromatography techniques, including column chromatography and high-performance liquid chromatography [12]. Identification of carotenoids can be achieved using spectroscopy methods such as nuclear magnetic resonance, infrared spectroscopy, and UV-Vis spectroscopy [13].

Chemically, carotenoids are a broad classification of isoprenoids characterized by extended hydrocarbon chains containing multiple single or double bonds and featuring cyclic or linear structures at their termini. Additionally, they can contain several functional groups in their chemical architecture, such as acetate, hydroxyl, epoxide, carboxylic, sulfate, and lactone groups [14]. Based on these features, carotenoids are classified into acyclic carotenes (e.g., lycopene and *ζ*-carotene), cyclic carotenes (e.g., *α*-carotene and *β*-carotene), carotenols (e.g., lutein and zeaxanthin), epoxycarotenoids (e.g., auroxanthin and luteoxanthin), crocetin glycosides (e.g., crocin), monoketo *β*-carotenes (e.g., echinenone), and 4,4′-diketo derivatives of *β*-carotene (e.g., canthaxanthin) [15].

Economically, the global market impact of carotenoids amounts to USD 1.44 billion. Among carotenoids, those with the highest industrial production comprehend astaxanthin, annatto, *β*-carotene, lutein, and lycopene, significant products used in dietary supplements, cosmetics, foods, and beverages [16]. Biologically, carotenoids are desirable compounds due to their ability to inhibit the growth of pathogenic Gram-positive (e.g., Staphylococcus aureus and Clostridioides difficile) and Gram-negative (e.g., Pseudomonas aeruginosa and Klebsiella oxytoca) strains [17]. Additionally, they regulate the levels of reactive oxygen species (ROS), scavenge free radicals, induce apoptosis of cancer cells, diminish the formation of new blood vessels, reduce the secretion of pro-inflammatory cytokines, and protect neurons against degradation [18].

In the clinical pipeline, carotenoids have been utilized to formulate multiple products to treat various disorders. Some of these drugs or supplements currently available in international markets include Pregvit^®^, Soriatane^®^, and Epuris^®^, which are indicated for reducing photosensitivity in patients, treating psoriasis, and managing severe recalcitrant nodular acne, respectively [19]. Due to their clinical safety and tolerability, carotenoids have been suggested as important adjuvant drugs in the treatment of COVID-19 [20], melanoma [21], breast cancer [22], diabetic retinopathy [23], and MetS [24].

In combating MetS, carotenoids have been shown to reduce the risk of diabetes and insulin resistance [25], downregulate blood pressure in hypertensive patients [26], reduce abdominal and epididymal adipose tissue weights [27], and slow the rise of blood glucose levels [28]. However, their broader applications are hindered by significant challenges, including low water solubility, high susceptibility to multifactorial degradation, high rates of oxidative degradation, and susceptibility to *trans*–*cis* isomerization [29].

In recent decades, micro- and nanotechnologies have emerged as active research fields to overcome significant challenges in the biomedical applications of bioactive natural products like carotenoids. Materials employed in both areas to improve these products’ performance, stability, and bioavailability include biopolymeric matrices such as chitosan, alginates, casein, or gelatin, as well as lipidic structures like liposomes or emulsions [30]. In the specific case of carotenoids, commonly utilized micro- and nanomaterials are gum Arabic [31], maltodextrin [32], fructo-oligosaccharides and inulin [33], and poly(vinylpyrrolidone) [34].

Given the need for a comprehensive review that addresses the potential of carotenoids as an alternative treatment in preventing and managing MetS, this review provides recent scientific evidence covering the extraction and characterization of carotenoids to their biological evaluation in vitro and in vivo using models that resemble conditions related to MetS disorders. In addition, the fundamentals of micro- and nanoencapsulation to protect the chemical integrity and preserve the therapeutic functionality of carotenoids are included. The search systems included Web of Science, PubMed, Springer, Wiley Online Library, and Google Scholar from 2020 to 2024. Keywords used to retrieve evidence were carotenoids, metabolic syndrome, microtechnology, nanotechnology, and structure–activity relationship. The number of publications related to the use of carotenoids against Mets and their number of citations are illustrated in Figure 1.

## 2. MetS: An Overview about Its Epidemiology, Oxidative Stress, and Treatment

### 2.1. Epidemiology

MetS encompasses various diseases, including hypertension, obesity, and diabetes mellitus. Figure 2 illustrates other diseases that are part of or associated with MetS, such as polycystic ovary syndrome, cardiovascular diseases (CVDs), dementia, non-alcoholic fatty liver disease (NAFLD), hypertension, and cancer. Discussing the epidemiology of MetS is challenging due to its association with various diseases, which can vary depending on age, gender, and ethnicity. However, it is documented that hypertension has been associated with over ~8.5 million deaths as of 2015, primarily in low- and middle-income countries [12]. In comparison, obesity is reported to affect ~650 million adults, particularly those in high- and middle-income countries [35], while diseases such as diabetes mellitus have affected ∼451 million people in recent decades [36].

### 2.2. Risk Factors

The development of MetS can be influenced by various factors, including oxidative stress, chronic inflammation, excessive alcohol and cigarette consumption, family history, and insufficient physical activity [37]. Although each factor plays a role in the progression of MetS-related diseases, the precise mechanisms remain elusive, with oxidative stress being the most frequently implicated. In patients diagnosed with MetS, oxidative stress may arise from mitochondrial and metabolic dysfunction caused by elevated levels of fatty acid or glucose and reduced insulin secretion or decreased levels of antioxidant enzymes in *β*-cells [38].

Free radicals such as ROS (H_2_O_2_, OH·, and ^1^O_2_), ONOO^−^, HNO_2_, and NO^∙^ can disrupt cell signaling pathways such as the polyol or protein kinase C pathways. This disruption leads to dysregulated mitochondrial energy levels and over-secretion of pro-inflammatory cytokines such as tumor necrosis factor-*α* (TNF-*α*), interleukins-2, -6, and -1*β* [39,40]. 

In the case of hypertension, recent studies have indicated that insulin resistance promotes increased lipid uptake by cells, such as myocardial cells, to meet energy demands, as glucose uptake is insufficient. This leads to more significant amounts of fatty acids for oxidation, potentially resulting in lipotoxicity. Consequently, ROS subsequently increases, leading to oxidative stress and damage to myocardial tissue, contributing to systolic dysfunction. Similarly, obese patients have been shown to have elevated levels of free fatty acids, increased secretion of pro-inflammatory cytokines (e.g., TNF-*α* and interleukin-6), and reduced activity of antioxidant enzymes (e.g., catalase and superoxide dismutase) [41]. Conversely, conditions like insulin resistance and type 2 diabetes (DM2) involve systemic inflammation and elevated ROS levels, resulting in various systemic damages such as extracellular matrix migration and deposition, endothelial dysfunction, cardiac hypertrophy and dysfunction, and fibrosis [42].

### 2.3. Treatment and Limitations

The treatment of MetS consists of administering antihypertensive drugs, lipid-lowering medications, antidiabetic medicines, and a combination of a healthy diet and physical activity. Current therapeutic regimens for hypertension include the administration of ARBs such as losartan [43], ACE inhibitors like lisinopril [44], or calcium channel blockers such as amlodipine [45].

Against diabetes, frequently administrated drugs include metformin, insulin, sulfonylureas such as glyburide [46], glucagon-like peptide-1 (GLP-1) receptor agonists like liraglutide [47], and sodium–glucose cotransporter 2 (SGLT2) inhibitors such as dapagliflozin [48]. In the case of anti-obesity drugs, common agents include lipase inhibitors such as orlistat [49], sympathomimetic appetite suppressants like phentermine [50], GLP-1 receptor agonists such as semaglutide [51], and serotonin receptor agonists like lorcaserin [52].

In clinical practice, administering ARBs, ACE inhibitors, antidiabetic agents, GLP-1 receptor agonists, or SGLT2 inhibitors can lead to limitations and adverse effects (AEs) that may affect patients’ well-being and prognosis. Co-administration of ACE inhibitors and ARBs has been associated with major cardiovascular events, all-cause mortality, gastrointestinal disorders, hypotension, angioedema, and hyperkalemia [53]. Similarly, the administration of GPL-1 agonists like semaglutide has been linked to serious AEs such as vomiting, pancreatitis, and diarrhea, while liraglutide use has resulted in upper abdominal pain [54]. Given these concerns, it is crucial to continue evaluating alternative treatments for MetS-related diseases.

## 3. Carotenoids: Sources, Extraction, Characterization, and Activities against MetS

### 3.1. Sources, Extraction, and Characterization

The advantages of carotenoids stem from their broad spectrum of biological activities and widespread presence in various sources.

Conventional extraction techniques, while effective, are associated with high energy consumption, long extraction periods, and low yield of bioactive products [55]. In contrast, green extraction methods, in line with the principles of green chemistry, offer an attractive alternative. These methods are versatile, allowing for decreased energy consumption and the use of new-generation solvents, while ensuring the obtention of high-quality products. Importantly, they can be performed with low volumes of green solvents such as ionic liquids, deep eutectic solvents, and natural deep eutectic solvents, which are non-volatile and biodegradable substances with the capacity to be recycled [56].

In recent years, green extraction methods have emerged as innovative techniques, offering a promising future for carotenoid extraction. These methods, such as ultrasound-assisted extraction (UAE), microwave-assisted extraction (MAE), and supercritical fluid extraction (SFE), can be optimized to yield high quantities of bioactive products while maintaining their eco-friendly nature and safety [57]. For instance, carotenoids such as 13-*β*-carotene, 9-*β*-carotene, and *γ*-carotene have been successfully extracted from peach palm (*Bactris gasipaes* Kunth), an Amazonian fruit rich in carbohydrates and oil by UAE [58]. Following the same technique, carotenoids have been obtained from orange peel utilizing olive oil as a solvent by controlling extraction time, temperature, and liquid-to-solid ratio [59].

Through UAE, carotenoids can be extracted from carrot pomace using hexane, acetone, or ethyl acetate while controlling experimental parameters such as liquid-to-solid ratio, ultrasonic power, and time [60]. In comparison with UAE, the efficacy of MAE to extract carotenoids from *Rhodotorula glutinis* using solvents of different polarity (dichloromethane, diethyl ether, dimethyl carbonate, and ethyl acetate) has been compared [57]. On the other hand, SFE with carbon dioxide (CO_2_) has been documented as a toxic waste-free process that does not cause the degradation of thermolabile compounds. The experimental setting of a CO_2_-SFE is depicted in Figure 3A, whereas Figure 3B illustrates the use of solvent extraction-based techniques for the recovery of carotenoids from palm-pressed fiber (PPF). CO_2_-SFE has been utilized to extract carotenoids from the cells of yeast *Rhodotorula* spp., strain ELP2022 under various operating parameters (pressure and temperature), and further separated by chromatography methods [61].

From microorganisms, carotenoids such as *β*-carotene can be extracted from yeasts like *Yarrowia lipolytica* using solid–liquid extraction with a solvent mixture (acetone, ethanol, and water), followed by solid–liquid–liquid extraction [64]. Alternatively, carotenoids such as astaxanthin have been extracted from genetically modified cell-derived cultures of microalgae such as *Haematococcus lacustris* [65], *Dunaliella salina*, *Chromochloris zofingiensis*, *Chloromonas krienitzii*, and *Sanguina nivaloides* [66]. Hydroxylated derivatives of astaxanthin, such as 2,2’-dihydroxy-astaxanthin and 2-hydroxy-astaxanthin, can be extracted with acetone from *Brevundimonas aurantiaca* M3d10, a bacterial strain found in wild olive flies [67,68]. Other genetically modified organisms to obtain astaxanthin encompass HEK293T cells, which are human embryonic kidney cells that have been proposed as an optimized efficient in vitro model for the biosynthesis of carotenoids [69]. Following extraction from these sources, carotenoids are chromatographically separated, quantified, and spectroscopically identified.

The separation of carotenoids is typically conducted using optimized column chromatography techniques such as high-performance liquid chromatography (HPLC), ultra-high-performance liquid chromatography (UHPLC), or ultra-performance liquid chromatography (UPLC). In contrast, to other chromatography methods, HPLC, UHPLC, and UPLC systems can be coupled to diode array detectors (DAD), photodiode array (PDA) detectors, and mass spectrometers (MS) to efficiently characterize the presence of bioactive compounds [70]. During the analysis of complex mixtures, it is relevant to consider the main features of each technique. For example, UHPLC and UPLC systems can operate at higher pressures, utilize columns with particle sizes less than ~2 μm, possess increased separation efficiency, and execute faster time analyses than HPLC systems. Despite their advantages, the instrumentation of UHPLC and UPLC systems is complex, and analysis of samples tends to be expensive [71].

For the evaluation of natural products, the use of DAD and PDA detectors is necessary since they enable the detection and identification of compounds based on their UV-Vis absorption spectra, which is a process with higher sensitivity compared to other detectors such as UV-Vis or refractive index detectors [72,73]. Even though DAD and PDA detectors are exploited in the investigation of bioactive compounds, it is crucial to mention that PDA detectors are preferred since they exhibit higher sensitivity, better signal-to-noise ratio, high data acquisition rates, higher spectral resolution, and advanced data processing [74]. In contrast, DAD detectors enable basic data processing features at slower scanning speeds with lower spectral resolution.

Adequate separation and further identification of carotenoids can be achieved by coupling HPLC, UHPLC, or UPLC systems with MS detectors. MS detectors are superior in identifying carotenoids, as they yield detailed information about the fragmentation patterns and molecular weight of molecules [75]. In contrast to DAD or PDA detectors, MS can be used with ionization techniques to quantify the presence of carotenoids among samples and confirm their identity [76]. Natural products can be identified through MS detectors by considering two or more stages of mass analysis within a mass spectrometer. The techniques derived from this process are known as tandem mass detection methods or MS/MS; a representative example includes a quadrupole–time-of-flight MS [77].

In the same context, column selection is another essential feature to consider when using chromatography methods. Column choice determines the successful separation of compounds [78]. In HPLC-based systems, columns exhibit different affinities for analytes in terms of polarity and charge, and they must possess compatible features with the composition of the mobile phase and parameters of the designed method, such as temperature and pressure [79]. Standard columns employed during HPLC analyses include C_18_ and C_30_ columns, which vary by their alkyl chain length, hydrophobicity, operating conditions, and selectivity. For the separation of carotenoids, representative chromatography columns comprehend platinum C_18_ 100A and betasil C_18_ columns [80].

To illustrate the extraction, characterization, and separation of carotenoids, Figure 4A represents the obtention of carotenoids from haloarchaeal strains and their characterization by UV-Vis spectroscopy, where it can be noticed that carotenoids exhibit major peaks at bands located at 385–522 nm [81]. Figure 4B depicts the HPLC analysis of lutein, zeaxanthin, and β-cryptoxanthin synthesized by a plant xanthophyll acyltransferase (XAT) by adding acyl donors [82]. Moreover, Figure 4C illustrates the extraction of carotenoids from *Rhodosporidium* sp., their identification by thin-layer chromatography (TLC) and HPLC, and evaluation against HEK293T cells and in silico modeling interactions with the vascular endothelial growth factor receptor 1 (VEGFR2) [83].

For the separation of carotenoids, an ultra-rapid resolution HPLC-DAD method was employed to detect free carotenoids like lutein, violaxanthin, and lycopene through liquid–liquid extraction from juices of mango, guava, pineapple, watermelon, and grape cultivars from Brazil [84]. This same technique has been utilized to analyze the presence of ketocarotenoids, such as astaxanthin, and 3-hydroxyechinenone in an acetone extract from *Balaustium muroum*, a free-living mite with a red body [85]. In contrast, fucoxanthin extracted from genetically modified cultures of *Nanofrustulum shiloi* was purified using preparative HPLC-DAD and subsequently analyzed by liquid chromatography–tandem mass spectrometry (LC-MS/MS) [86]. Table 1 provides an overview of carotenoids extracted from different sources and the techniques used for their separation and characterization.

Spectroscopy and microscopy techniques have been widely used to characterize carotenoids in laboratory settings and clinical studies. For example, FTIR spectroscopy was employed to analyze the functional groups present in flour derived from the peels of passion fruit (*Passiflora edulis f.* flavicarpa), known for its natural abundance of carotenoids [101]. In another study, nuclear magnetic resonance (NMR) was utilized to investigate changes in the chemical structure of carotenoids like cryptoxanthin and zeaxanthin from chili oil subjected to different frying temperatures: 30, 150, 170, and 190 °C [102]. In cross-sectional studies, spectroscopy approaches such as Raman spectroscopy have been used to monitor Korean adults’ carotenoid intake [103]. Furthermore, in parallel controlled trials, pressure-mediated reflection spectroscopy has been applied to detect changes in skin carotenoid levels, such as lycopene, lutein, phytoene, and phytofluene, in adults diagnosed with obesity [104]. Regarding the use of microscopy methods, fluorescence lifetime imaging microscopy has been implemented to analyze and correlate carotenogenesis events among cellular structures from *Bracteacoccus aggregatus* cultures [105].

### 3.2. Activities against MetS

Against hypertension, dietary carotenoids have been shown to reduce blood pressure and atherosclerosis in patients diagnosed with CVDs. Lycopene has been implicated in exerting preventive effects during the treatment of CVDs. It can downregulate the generation of ROS, modulate the activation of NF-κB, a major pro-inflammatory pathway, and thereby regulate the expression of cell adhesion factors and vascular permeability [106]. Similarly, high serum levels of *α*-carotene, *β*-carotene, *β*-cryptoxanthin, lycopene, lutein, and zeaxanthin have been associated with a decreased risk of mortality caused by CVDs among adults in the USA [26]. Moreover, it has been revealed that low serum lycopene concentrations are linked to an increased risk of death from CVDs, while moderate serum levels are associated with protective effects against cardiovascular death in recent cross-sectional survey studies [107].

The consideration of carotenoids as useful bioactive nature products against obesity is attributed to their capacity to affect adipogenesis, modulate metabolic capacity, and decrease the release of inflammatory products [108]. In C57BL/6J obese mice models, treatment with lutein was found to decrease epididymal and abdominal adipose tissue weights, reduce serum cholesterol, low-density lipoprotein cholesterol (LDC-C) concentration, hepatic triglycerides, and cholesterol, as well as blood glucose levels [27]. These effects were also analyzed in combination with orlistat. Similarly, C57BL/6N mice treated with carotenoids from red paprika (e.g., *α*-carotene, *β*-carotene, capsorubin, lutein, and zeaxanthin) showed enhanced endurance exercise capacity, increased muscle weights of limbs, and diminished intramyocellular fat accumulation, prevented the degradation of muscle protein through the activation of the mTOR pathway, and enhanced mitochondrial dysfunction [109].

The effect of carotenoids on diabetic patients or clinical models has been an active research field over the last decades. For instance, it was demonstrated that treatment with astaxanthin decreased the levels of low-density lipoproteins and cholesterol while improving whole-body glucose disposal and insulin sensitivity without causing apparent AEs among prediabetic patients [110]. Similarly, fucoxanthin, an algae-specific xanthophyll, was documented through a randomized, double-blind placebo-controlled clinical trial to increase the total and first phase of insulin secretion while decreasing body weight, body mass index, waist circumference, systolic blood pressure, diastolic blood pressure, and triglyceride levels among patients with MetS [111].

Against other diseases that constitute MetS, such as cancer, high circulating levels of carotenoids such as *β*-carotene and vitamin A have been demonstrated to reduce breast cancer risk while promoting the production of metabolites involved in redox balance, immune response regulation, and synthesis of macromolecules. This was reported through metabolomics analyses [112]. In the case of disorders that arise from cognitive decline, such as dementia, *α*- and *β*-carotene, lutein, and zeaxanthin have been associated with improved cognitive functionality [113]. On the other hand, treatment with lycopene has been demonstrated to prevent the development of NAFLD in C57BL/6J mice by suppressing the hepatic NFκ-B/NLRP3 inflammasome pathway [114]. In contrast, treatment with zeaxanthin has been shown to decrease the generation of ROS, inhibit ferroptosis, enhance mitochondrial dysfunction, and diminish lipid peroxidation in free fatty acid-induced HepG2 cells. These effects are promising for further validating the role of carotenoids in NAFLD intervention [115].

## 4. Encapsulation of Carotenoids

Encapsulation is the process whereby bioactive substances and their physicochemical features are protected from environmental factors (e.g., temperature, light, and moisture) or physiological changes (e.g., transport mechanisms, enzyme degradation, and binding proteins) by trapping them within matrices synthesized from different materials [116]. As illustrated in Figure 5A, the release of therapeutic cargo from micro- and nanostructures can be triggered by external and internal stimuli such as heat, ultrasound, light, magnetic field alterations, pH, redox conditions, and enzyme activity. In contrast, via linker chains, their uptake by tissues or cells can be facilitated by attaching natural or synthetic targeting moieties such as enzymes, antibodies, folic acid, or polyethylene glycol [117]. This event is also depicted in Figure 5A.

Microstructures range from 1 to 1000 μm, and they can be manufactured from synthetic, semi-synthetic, and natural components through physicochemical routes, such as solvent evaporation, spray-drying, and anionic polymerization methods, respectively [118,119,120]. As represented in Figure 5C, microparticles are convenient as they can be manipulated to formulate solid, semi-solid, and liquid products that can be administered orally, intravenously, intramuscularly, or subcutaneously to successfully deliver and enhance the efficacy and bioavailability of one or more drugs [121].

In contrast, nanomaterials are conventionally found from 10 to 500 nm and are frequently exploited for the development of drug delivery systems [122], bioactive structures [123], and platforms for disease diagnosis [124]. Like microparticles, nanomaterials are synthesized from organic and inorganic substances through top-down (e.g., laser ablation, lithography, and vapor deposition) and bottom-up (e.g., flame spraying, sol–gel, and micro-emulsion) approaches [125]. They can be administered through various routes, such as intranasal, intraventricular, and intraparenchymal [126]. The administration routes of nanomaterials are depicted in Figure 5C. However, the difference between both categories lies in the fact that nanomaterials are preferred because they can mediate molecular and cellular phenomena, cross endothelial cells of major anatomical structures, and interact with receptors or recognize ligands [126,127].

Given their versatility in the encapsulation of bioactive products, nanoparticles are frequently synthesized to improve the therapeutic activities of carotenoids. As noted in Figure 6A, carbon nanoparticles containing carotenoids can be used to develop multifunctional systems with antiproliferative effects towards melanoma and breast cancer cell lines and 3D imaging capacity in skin tissues [128]. As mentioned, these events can arise if nanoparticles are tailored with molecules, which, in the case of Figure 6A, were attributed to the incorporation of phospholipids onto the surface of carbon nanoparticles. The functionalization of carbon nanoparticles resulted from a complex process that included nucleation, passivation, membrane-freeze, and sonication stages. This can be observed in Figure 6A.

The physical features such as size and morphology dictate their fate in organisms. Dynamic light scattering (DLS) and transmission electron microscopy (TEM) are frequently used to determine these parameters. As observed in Figure 6B, DLS analyses of cationic liposomes loaded with carotenoids (lutein and *β*-carotene) can reveal their size distribution patterns and demonstrate the presence of structures with diverse sizes. In the case of TEM, it is used to determine the morphological arrangement of micro- or nanostructures. As observed in Figure 6C, cationic liposomes loaded with lutein and *β*-carotene have been demonstrated to adopt oval shapes and small aggregates.

The encapsulation of carotenoids is necessary as they tend to be rapidly degraded upon exposure to heat, oxygen, and light. Other factors that alter their stability include transition metals and radical species [130]. Various micromaterials have been used to prevent the degradation of carotenoids or their isomers; for example, supramolecular oligosaccharides such as cyclodextrins [131], soluble fibers such as pectin [132], and protein residues from cereal grains such as barley [133].

There is scientific evidence regarding using micro- and nanomaterials to treat, in vitro or in vivo. However, Table 2 compiles recent studies where micro- and nanomaterials have been used to encapsulate carotenoids and exhibit potential functional activities to decrease or mitigate molecular disorders related to MetS.

In comparison, common nanomaterials used to entrap carotenoids include polymer-coated liposomes [153], gelatin-based nanoparticles (NPs) [154], and hydrogels synthesized from starch blends [155]. Scientific evidence regarding common carotenoids entrapped into micro- or nanomaterials to treat diseases related to MetS is limited, but it is compiled in the following sections. Table 3 provides examples of nanomaterials used to entrap carotenoids and their biological properties.

### 4.1. Astaxanthin

Astaxanthin is a carotenoid composed of forty carbon atoms, fifty-two hydrogen atoms, and four oxygen atoms (C_40_H_52_O_4_). Modern advances have involved encapsulating astaxanthin in micro- and nanomaterials. For instance, NPs prepared with astaxanthin and anthocyanins have demonstrated enhanced activity of superoxidase dismutase (SOD), glutathione peroxidase (GSH-PX), and catalase while reducing fat content, lipofuscin accumulation, and ROS levels on high-fat *Caenorhabditis elegans* cultures. These properties are appealing for the treatment of DM2 [167]. In obesity, emulsion-based delivery systems containing astaxanthin have been shown to reduce body fat accumulation, hepatic fatty acid, and hepatic lipid levels in obese mice models. These effects are attributed to the high oral absorbability of the prepared system [168].

In other studies, nanocarriers containing astaxanthin, formulated with galactose, whey protein isolate, and triphenyl-phosphonium, have been shown to target mitochondria in steatotic HepG2 cells exhibiting potent anti-adipogenic and antioxidative activities. Moreover, these carriers altered blood lipid levels in NAFLD mouse models and reduced liver lipid accumulation [169]. In rat models treated with CCl_4_, astaxanthin NPs prepared with lecithin ameliorated hepatic damage and decreased plasma biomarker levels such as aminotransferase and aspartate aminotransferase [170]. Against potential neurological disorders, NPs or nanostructured lipid carriers containing astaxanthin demonstrated protective effects against MPP^+^ (an active metabolite that mimics Parkinson’s disease), ferric iron, and tert-butyl hydroperoxide in SH-SY5Y cells [171].

On the other hand, microencapsulation has also been explored with this carotenoid. Spray drying utilizing gum Arabic has shown promising effects, including increased solubility, stability, and improved bioavailability in the simulated gastrointestinal tract (GIT). It was observed that variations in inlet and outlet temperatures could impact the encapsulation efficiency, with higher microencapsulation efficiency noted at higher inlet temperatures ranging from 180 to 190 °C. This is attributed to the accelerated formation rate of microcapsules under these conditions [134]. In another study, multilayer O/W emulsions were formed using ι-carrageenan, chitosan, lupine protein isolate, and sunflower oil, followed by spray drying. The effects of this technique include an increase in astaxanthin retention and storage stability, achieved by providing protection against oxidative damage of astaxanthin. It was observed that O/W emulsions exhibit greater physical stability in two- and three-membrane systems compared to single-membrane systems. Additionally, a greater solubility in water was noted [135]. On the other hand, ionic gelation can be employed using low-methoxyl pectin, chitosan, and alginate to form microcapsules. This microencapsulation method improves particle sphericity and limits oil oxidation during formulation. Increased thermal stability and bioavailability in simulated GIT were also observed. The authors concluded that the appropriate ratio of alginate to pectin could result in ideal encapsulation efficiency and bioaccessibility, with these values depending on the concentrations of the polymers [132].

### 4.2. β-Carotene

The evidence regarding micro- or nanomaterials prepared with isolated or commercially available *β*-carotene (C_40_H_56_) is limited. Previous studies have shown that NPs loaded with carotenoid-rich extracts from cantaloupe melon enhanced the levels of hepatic retinol in Wistar rats with obesity without affecting tissue integrity [172]. On the other hand, *β*-carotene-based nanoemulsions combined with quercetin have been reported to protect hepatic and pancreatic *β*-cells against damage, reduce blood sugar levels, influence oral glucose tolerance, control body weight, decrease tissue damage markers such as aspartate aminotransferase and alanine aminotransferase, and enhance SOD levels in diabetic albino Wistar rats [173]. In another study, calcium–alginate beads were fabricated using the extrusion process to enhance the stability and preserve the DPPH antioxidant activity of olive oil, which was enriched with carotenoids such as *β*-carotene [59].

Similarly, protein-based nanoemulsions have demonstrated the capacity to increase the bioavailability of *β*-carotene and retinol using in vitro (simulated gastrointestinal digestion) and in vivo (female Sprague-Dawley rats) models [174]. Other formulations to enhance the bioavailability of *β*-carotene have been observed with emulsion formed with insoluble soybean, protein hydrolysate, and xanthan gum, which exhibited enhanced thermal, ionic, and pH stability while promoting the bioavailability of this compound under simulated gastric conditions [152].

The microencapsulation of this carotenoid with gum Arabic, maltodextrin, modified starch, and whey protein was tested using the spray drying technique. The results showed an increased half-life of *β*-carotene and a decreased fraction of encapsulated *β*-carotene. It was also observed that the agglomeration of the microparticles positively affected their solubility in water [149]. Another technique tested with this carotenoid was complex coacervation with Amaranth carboxymethyl starch and lactoferrin. Enhanced encapsulation efficiency, thermal protection at 50 °C, and photolytic stability against UV radiation were observed. Likewise, a high intestinal release in oily matrices was observed, along with carotenoid protection against gastrointestinal conditions [150].

Likewise, ionotropic gelation was tested using sodium caseinate and κ-carrageenan as materials. High encapsulation efficiency, *β*-carotene retention, and low viscosity were observed. Some losses of the carotenoid were noted due to probable escapes of oil droplets during the formation of the microparticles in the gelation process. Additionally, it was observed that the polysaccharide concentration influenced the morphology of the particles, while the pH did not significantly affect characteristics such as morphology and encapsulation efficiency [151]. On the other hand, aggregated insoluble soybean protein hydrolysate and xanthan gum were used to create a W/O/W emulsion. An increased encapsulation efficiency (70.56 ± 0.06%) and suitable pH values (5 to 11), ionic stability, and thermal stability (25 to 85 °C) were observed. Similarly, the technique enhanced bioavailability in simulated GIT. An advantage of the technique is its ability to protect and deliver both hydrophilic and hydrophobic components, as is the case of this carotenoid [152].

### 4.3. Crocin

The effect of microencapsulation on crocin (C_44_H_64_O_24_) has been tested using spray drying with gelatin as an ingredient. A direct proportionality was noted between the gelatin concentration and particle size, making obtaining smaller microcapsules with lower gelatin concentrations possible. As a result, increased encapsulation efficiency and preserved crocin physicochemical properties were observed [146]. On the other hand, ionotropic gelation was used with alginate as the primary material. This resulted in high encapsulation efficiency and fast kinetics release for the microparticles obtained. Crocin, unlike the carotenoids discussed above, is soluble in water. However, with microencapsulation, it was observed that the release of the molecule in an aqueous environment could be controlled, presenting an advantage of using this technique [147]. In another study, the ionotropic gelation technique was used with chitosan, gelatin, and oxidized alginate as materials. The observed effects of the hydrogel were high encapsulation efficiency, sustained crocin release, and superior mucoadhesive strength. It was also proven that with a lower concentration of chitosan, there was a more efficient gelation process since the mixture’s viscosity was reduced, and the solubility of crocin was favored [148].

### 4.4. Fucoxanthin

Fucoxanthin is a carotenoid with forty-two carbon, fifty-eight hydrogen, and six oxygen atoms (C_42_H_58_O_6_). Over the last decades, fucoxanthin has been entrapped into structures of different materials. Recently, it was documented that fucoxanthin and zein hydrolysate encapsulated into NPs diminished liver pathology, decreased blood glucose levels, influenced the expression of genes (e.g., GLP-1, GLUT2, and PI3K) related to glucose metabolism, and improved the restoration of intestinal microorganisms correlated to the gut microbiota in C57BL/6J mice [175]. In other studies, fucoxanthin into hydrolyzed zein nanocomplexes has been reported to decrease and restore fasting blood glucose levels in DM2 mice (C57BL/6) while affecting oxidative stress, enhancing SOD and GSH-PX levels, upregulating GLUT2 expression, promoting repair of hepatocyte and pancreatic β-cell damage, and regulating glycogen synthesis in the same model [176].

Using gums and proteins of natural origin, gum Arabic/gelatin microcapsules containing fucoxanthin and incorporated into alginate hydrogel beads decreased body weight. It lowered blood lipid content (high-density lipoprotein cholesterol, total cholesterol, and low-density lipoprotein cholesterol) and oxidative stress levels in specific-pathogen-free (SPF) grade Kunming (KM) mice models after oral administration. In the same study, treatment with microcapsules was biocompatible with L929 cells [138]. On the other hand, microencapsulation with maltodextrin, soy lecithin, and gum Arabic through spray drying and freeze drying has resulted in microcapsules with high encapsulation efficiency and increased bioavailability, as well as ABTS^•+^ scavenging activity.

Furthermore, it was observed that the microcapsules obtained by freeze drying had a higher encapsulation efficiency than those obtained by spray drying, resulting in a higher concentration of the carotenoid and increased observed activity [136]. In another study, microencapsulation through sequential coating modification using maltodextrin and gum Arabic enhanced encapsulation efficiency and fucoxanthin stability. The authors describe that this technique increased stability against damage by light and heat, making it viable for use in several industries where fucoxanthin is exposed to these conditions [137]. Additionally, the efficacy of complex carriers was evaluated using gum Arabic, gelatin, and alginate hydrogel as materials, and their in vitro and in vivo activity was tested. Increased bioavailability and fucoxanthin protection against simulated gastric fluid (SGF) were observed in in vitro tests. On the other hand, in in vivo tests, a decrease in blood lipids and oxidative stress levels was demonstrated after oral administration. Furthermore, fucoxanthin accumulation in the intestinal tract also decreased weight in rats fed a high-lipid diet [138].

On the other hand, nanocomplexes loaded with fucoxanthin composed of gelatin and chitosan oligosaccharides have been shown to reduce the generation of ROS, decrease mitochondrial damage upon exposure to H_2_O_2_, inhibit apoptosis, and regulate lipid metabolism in L02 cells [177]. In another study, fucoxanthin isolated from *Sargassum wightii* Greville exhibited antidiabetic and antihypertensive properties by reducing oxidative stress, preserving pancreatic tissue morphology, and reducing hyperglycemic condition [178].

### 4.5. Lycopene

Lycopene (C_40_H_56_), a carotenoid commonly present in tomato skin, is susceptible to environmental factors such as light and oxygen degradation. In recent studies, lycopene was incorporated into liposomes loaded with tobramycin to develop a multifunctional antibiotic hydrogel. This hydrogel demonstrated the ability to inhibit bacterial proliferation, promote cell migration and angiogenesis, stimulate collagen deposition, and enhance wound healing in an infected diabetic wound model in rats [179].

In another study, lycopene was initially encapsulated within liposomes and then integrated into a porous chitosan microgel along with nicotinamide mononucleotide. This formulation was found to exert protective effects against acute liver injury in C547BL/6 mice and regulate gut microbiota [180]. The protective effect on the liver was attributed to the microgel’s ability to inhibit the TLR4/NF-κB signaling pathway and interact with the TLR4/MD2 complex, reducing inflammatory and oxidative stress responses. Similarly, sea buckthorn juices containing lycopene were entrapped within microspheres derived from sodium alginate and κ-carrageenan. These microspheres were reported to regulate the release and control of blood glucose levels by inhibiting the activity of α-amylase and α-glucosidase, respectively [141].

Regarding nanomaterials, lycopene loaded into H-ferritin nanocages, along with triphenylphosphonium, has been reported as a novel targeted therapy for intracerebral and intra-neuronal treatment to address memory loss and neuronal dysfunction. This formulation can regulate mitochondrial function in nerve cells, preserve mitochondrial morphology, prevent memory decline, and promote synaptic plasticity in aging mice [181]. Similarly, treatment with sequence-targeted lycopene nanodots encapsulated within H-ferritin nanocages protected mitochondrial functionality by enhancing ROS scavenging activity and promoting neural enrichment and mitochondrial regulation in C57/BL mice [182]. Additionally, these nanodots exhibited pro-survival mitophagy activities and induced the degradation of pathogenic α-synuclein.

Since microencapsulation has been another method of protecting molecules with specific bioactivity, experiments have been reported on preparing lycopene microcapsules. One study evaluated microencapsulation with gum Arabic and inulin using spray drying. The microcapsules demonstrated high lycopene release in simulated gastric fluid. However, a decrease in antioxidant activity was observed, probably due to exposure of the carotenoid-containing solutions to environmental factors such as oxygen and light during the microcapsule preparation process. Furthermore, the encapsulated extract showed more excellent stability against light and oxygen than the pure extract when both were stored for 27 days [139,140]. On the other hand, lycopene microencapsulation was achieved through ionotropic gelation using alginate and *κ*-carrageenan in another experiment. In a separate experiment, whey proteins and acacia gum were utilized to produce microcapsules through complex coacervation and freeze drying. The observed effect was increased antioxidant activity, and in vitro α-amylase and α-glucosidase inhibitory activity was demonstrated, with a greater effect on α-amylase. The observed enzymatic activity can be categorized as a preventive effect against DM2 and MetS [142].

### 4.6. Lutein

Lutein is a linear xanthophyll with forty carbon atoms, fifty-six hydrogen atoms, and two oxygens (C_40_H_56_O_2_). In an innovative study, lutein was encapsulated into chitosan-coated liposomes and further reported to decrease the levels of ROS and enhance the performance of SOD and GHS-PX in vitro. In the same study, treatment with lutein diminished total cholesterol and triglycerides levels, body weight, and fat accumulation in epididymal adipose tissue and liver of high-fat diet C57BL/6J mice [183]. Similarly, lutein-coated poly (lactic-co-glycolic acid) NPs coated with macrophage membranes inhibited angiotensin II-induced primary cardiac fibroblast proliferation, improved cardiac function and structure, and regulated cardiac fibrosis without causing significant toxicity in C57BL/6 male mice [184].

Regarding microencapsulation, inulin and modified starch were used to produce powdered microcapsules via spray drying, enhancing lutein stability and resistance to thermal degradation. In recent studies, the spray drying technique was employed with citric acid-esterified potato starch and whey protein as the main components. Microcapsules produced via this method exhibited increased encapsulation efficiency, embedding effect, lutein aqueous solubility, and thermal resistance. It was observed that smaller microcapsules had better solubility in water. Therefore, it can be inferred that the choice of materials can influence the effectiveness of the encapsulation technique, even when the same spray drying method is utilized [144]. Similarly, sodium caseinate and alginate were utilized to microencapsulate lutein through electrostatic complexation. The observed effects included reduced lutein decomposition during storage compared to carotenoid emulsions. It was evident that degradation due to oxidative damage was mitigated by the protective barrier, preventing free radicals from affecting the lutein molecule. Furthermore, increased FFA release and lutein bioaccessibility were observed in simulated GIT [145].

## 5. Conclusions

Carotenoids, ubiquitous bioactive compounds, have been extensively utilized in formulating products significantly impacting global health care and economic sectors. Evidence regarding the prevalence of MetS primarily focuses on patients diagnosed with diabetes, obesity, or cardiovascular diseases. Present therapeutic approaches are tailored to specific MetS-related conditions, often presenting due to the potential for AEs.

In recent decades, the utilization of carotenoids has emerged as an appealing option for mitigating or preventing the onset of MetS. This is attributed to their advantageous role in regulating lipid metabolism, reducing blood sugar levels, augmenting the activity of antioxidant enzymes, and averting tissue and cellular damage. Nonetheless, their application is impeded by their vulnerability to degradation caused by environmental and physiological factors.

From the available literature, it is evident that carotenoids are primarily sourced from biological materials, with the UAE method being the preferred extraction technique. However, their characterization using spectroscopy techniques is still limited, which poses a challenge in gaining a comprehensive understanding of their chemical features and biological performance. This limitation opens avenues for innovative research. Moreover, the use of micro- and nanomaterials to address these limitations and their evaluation in models that simulate MetS conditions is a relatively new approach, with only a few studies meeting these criteria. Furthermore, in cases where studies focus on the use of carotenoids and materials with micro- or nanostructured arrangements, they often struggle to establish clear relationships between morphology or surface charge and the activity of the entrapped substance against MetS models. This highlights the need for further investigation in this area to elucidate the mechanisms underlying their therapeutic potential.

## Figures and Tables

**Figure 1 plants-13-01584-f001:**
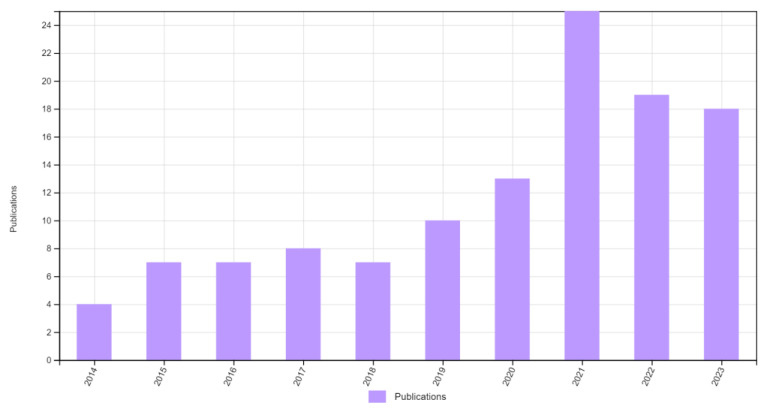
Time distribution of published papers related to the investigation of carotenoids against MetS.

**Figure 2 plants-13-01584-f002:**
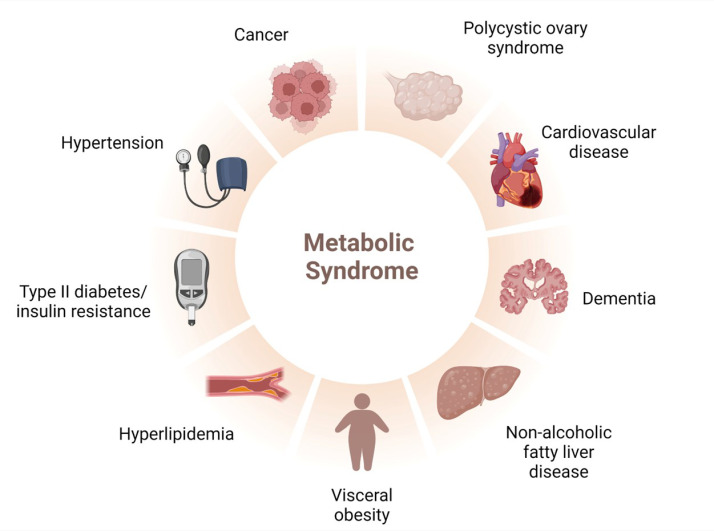
Cluster of diseases constituting metabolic syndrome. Adapted from BioRender.com.

**Figure 3 plants-13-01584-f003:**
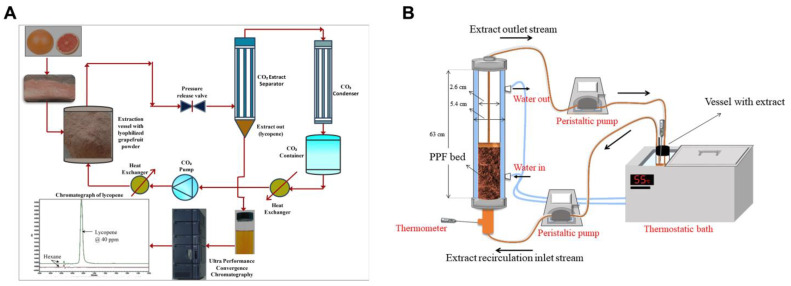
(**A**) Representation of a CO_2_-SFE system for the extraction of lycopene from ripe grapefruit endocarp; reprinted and adapted with permission from [62]. (**B**) Solvent extraction system of carotenoids from palm-pressed fiber (PFF) utilizing a peristaltic pump, thermostatic bath, and thermometer; reprinted and adapted with permission from [63].

**Figure 4 plants-13-01584-f004:**
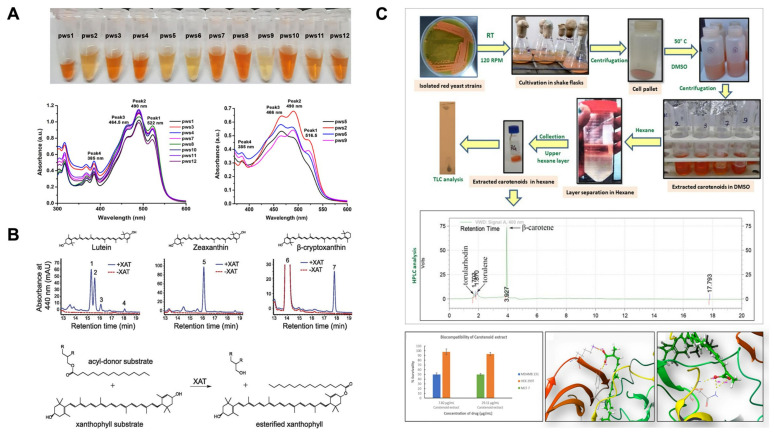
(**A**) Carotenoids isolated from twelve strains of haloarchaeal strains categorized into two main categories (yellow and orange) and their absorbance profiles obtained by UV-Vis spectroscopy analyses; figure was reprinted and adapted with permission from [81]. (**B**) HPLC analysis of lutein, zeaxanthin, and β-cryptoxanthin synthesized by a plant xanthophyll acyltransferase (XAT) by adding acyl donors [82]. (**C**) Cultivation of red yeast strains, extraction, separation with hexane, and chromatography analysis of carotenoids by thin-layer chromatography (TLC) and HPLC, and their effect in HEK293T cells and in silico simulation; figure was reprinted and adapted with permission from [83].

**Figure 5 plants-13-01584-f005:**
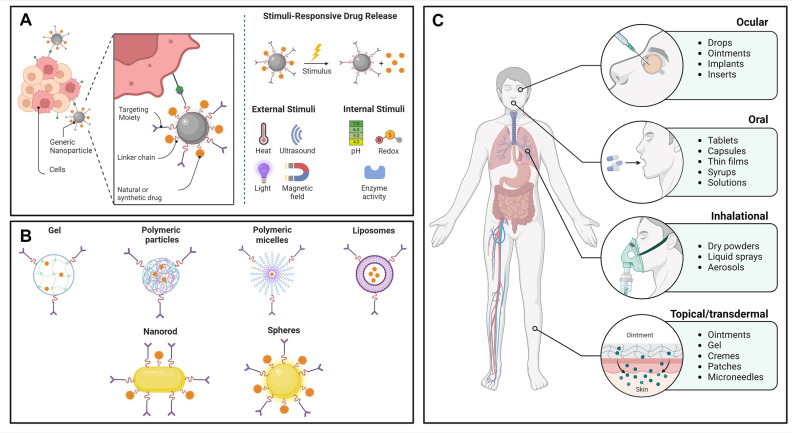
(**A**) Schematic representation of functionalization of micro- or nanomaterials with natural and synthetic molecules to improve their targeting capacity and release of therapeutic cargo through external (heat, ultrasound, light, and magnetic field) and internal stimuli (pH, redox conditions, and enzyme activity). (**B**) Classification of materials as gels, polymeric particles, polymeric micelles, liposomes, nanorods, and spheres. (**C**) Administration routes of micro- and nanomaterials via ocular (drops, ointments, implants, and inserts), oral (tablets, capsules, thin films, syrups, and solutions), inhalational (dry powders, liquid sprays, and aerosols), and topical/transdermal (ointments, gel, cremes, patches, and microneedles). Adapted from BioRender.com.

**Figure 6 plants-13-01584-f006:**
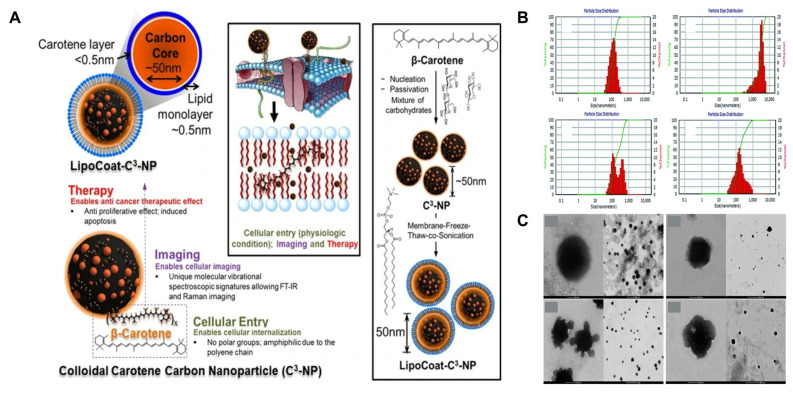
(**A**) Development of carbon nanoparticles (C^3^-NP) containing carotenoids and coated with phospholipids to enhance their antiproliferative effect against melanoma and cancer cell lines, imaging capacity, and cellular entry; figure was reprinted and adapted with permission from [128]. (**B**) Dynamic light scattering (DLS) analysis of cationic liposomes prepared with β-carotene; figure was reprinted and adapted with permission from [129]. (**C**) Transmission electron microscopy (TEM) analysis of cationic liposomes loaded with lutein or *β*-carotene; figure was reprinted and adapted with permission from [129].

**Table 1 plants-13-01584-t001:** Sources, extraction techniques, separation methods, and identification approaches of carotenoids.

Carotenoid	Source	Extraction Technique	Separation Method	Identification Approach	References
	*Haematococcus pluvialis*	Liquid–liquid extraction	HPLC	UV-Vis spectroscopy FTIR spectroscopy	[87]
Astaxanthin	*Corynebacterium glutamicum*	Maceration	HPLC	N.I.	[88]
	*Phaffia rhodozyma*	Solid–liquid extraction	N.I.	UV-Vis spectroscopy	[89]
	*Tisochrysis lutea*	Solid–liquid US-assisted extraction	HPLC and CPC	N.I.	[90]
Fucoxanthin	*Undaria pinnatifida*	Heat extraction	HPLC	UV-Vis spectroscopy	[91]
	Tomato (*Solanum lycopersicum*)	Enzyme-assisted extraction	UHPLC	N.I.	[92]
	Carrot (*Daucus carota*)	Organic solvent extraction	N.I.	UV-Vis spectroscopy	[93]
Lycopene	*Elaeagnus umbellata*	UAE	UHPLC	UV-Vis spectroscopy FTIR spectroscopy MS	[94]
	Red papaya	SC-CO_2_	N.I.	N.I.	[95]
	Pistachio waste	Soxhlet extraction	LC-MS/MS	MS	[96]
Lutein	Fruit juices	Liquid–liquid extraction	HPLC	N.I.	[84]
	Marigold flowers	Surfactant-based ATPS extraction	N.I.	UV-Vis spectroscopy	[97]
	*Lycium barbarum*	Liquid–liquid extraction	HPLC HSCCC	N.I.	[98]
Zeaxanthin	Dried corn silk	Solid–liquid extraction	HPLC CC	UV-Vis spectroscopy FTIR spectroscopy NMR spectroscopy	[99]
	*Chlorella*	PLE	HPLC	UV-Vis spectroscopy	[100]

Abbreviations: SC-CO_2_, supercritical carbon dioxide; ATPS, two-phase system; PLE, pressurized liquid extraction; HPLC, high-performance liquid chromatography; CPC, centrifugal partition chromatography; UHPLC, ultra-high-performance liquid chromatography (HPLC), liquid chromatography–tandem mass spectrometry; HSCCCU, high-speed countercurrent chromatography; UV-Vis, ultraviolet–visible; FTIR, Fourier transform infrared spectroscopy; MS, mass spectrometry; N.I., not indicated.

**Table 2 plants-13-01584-t002:** Examples of microencapsulation systems of carotenoids and the observed activities.

Carotenoid	Encapsulation Technique	Raw Materials	Observed Activities	References
Astaxanthin	Spray drying	Gum Arabic	Increased solubility, stability, and enhanced bioavailability in simulated GIT.	[134]
	Multilayer O/W emulsion and spray drying	*ι*-carrageenan, chitosan, lupin protein isolate, and sunflower oil	Augmented astaxanthin retention, storage stability, and water solubility.	[135]
	Ionic gelation	Low-methoxyl pectin, chitosan, and alginate	Improved particle sphericity and limited oil oxidation during formulation. Increased thermal stability and bioavailability in simulated GIT.	[132]
Fucoxanthin	Spray drying and freeze drying	Maltodextrin, soy lecithin, and gum Arabic	Microcapsules exhibited high encapsulation efficiency, increased bioavailability, and ABTS•^+^ scavenging activity.	[136]
	Sequential coating modification	Maltodextrin and gum Arabic	Enhanced encapsulation efficiency and fucoxanthin stability.	[137]
	Complex carriers	Gum Arabic, gelatin, and alginate hydrogel	In vitro increased bioavailability and fucoxanthin protection against SGF. In vivo oral administration demonstrated lowering the blood lipid and oxidative stress levels.	[138]
Lycopene	Spray drying	Gum Arabic and inulin	Microcapsules demonstrated high lycopene release in simulated gastric fluid.	[139,140]
	Ionotropic gelation	Alginate and *κ*-carrageenan	Microspheres exhibited in vitro *α*-amylase and *α*-glucosidase inhibitory activity.	[141]
	Complex coacervation and freeze drying	Whey proteins and acacia gum	Formulation increased antioxidant activity and showed in vitro *α*-amylase and *α*-glucosidase inhibitory activity.	[142]
Lutein	Spray drying	Inulin and modified starch	Microencapsulated powders significantly increased lutein stability and thermal degradation resistance.	[143]
		Citric acid-esterified potato starch and whey protein	Increased encapsulation efficiency, embedding effect, lutein aqueous solubility, and thermal resistance.	[144]
	Electrostatic complexation	Sodium caseinate and sodium alginate	Reduced lutein decomposition during storage and increased FFA release and lutein bio accessibility in a simulated GIT.	[145]
Crocin	Spray drying	Gelatin	Increased encapsulation efficiency and preserved crocin physicochemical properties.	[146]
	Ionotropic gelation	Alginate	High encapsulation efficiency and fast kinetics release.	[147]
		Chitosan, gelatin, and oxidized alginate	Hydrogel exhibited high encapsulation efficiency, sustained crocin release, and superior mucoadhesive strength.	[148]
*β*-carotene	Spray drying	Gum Arabic, maltodextrin, modified starch, and whey protein	Increased *β*-carotene half-life and decreased the *β*-carotene encapsulated fraction.	[149]
	Complex coacervation	Amaranth carboxymethyl starch and lactoferrin	Enhanced encapsulation efficiency and thermal and photolytic stability. High intestinal release in oily matrices.	[150]
	Ionotropic gelation	Sodium caseinate and *κ*-carrageenan	High encapsulation efficiency and *β*-carotene retention. Low viscosity.	[151]
	W/O/W emulsion	Aggregated insoluble soybean protein hydrolysate and xanthan gum	Increased encapsulation efficiency and good pH, ionic, and thermal stability. Enhanced bioavailability in simulated GIT.	[152]

Abbreviations: O/W, oil/water; GIT, gastrointestinal tract; ABTS•^+^, 2,2′-Azino-bis(3-ethylbenzothiazoline-6-sulfonic acid) diammonium salt radical cation; SGF, simulated gastric fluid; FFA, free fatty acids; W/O/W; water/oil/water.

**Table 3 plants-13-01584-t003:** Examples of nanoencapsulation systems of carotenoids and the observed activities.

Carotenoid	Encapsulation System	Raw Materials	Observed Activities	References
Astaxanthin	Nanocapsules	Formaldehyde and lysine	Nanocapsules reduced the production of H_2_O_2_ and maintain mitochondrial membrane potential. Nanocapsules exhibited stability against high temperatures, pH, and UV radiation.	[156]
	NPs	Chitosan and sodium triphosphate	NPs exhibited sustained in vitro release in simulated gastric and intestinal conditions. NPs loaded with astaxanthin executed prolonged residence, time levels, and antioxidant activities in Sprague-Dawley rats.	[157]
	Nanoemulsion	Soybean protein isolate and sodium alginate	Nanoemulsion presented scavenging activity against H_2_O_2_ and DPPH radicals. Nanoemulsion exhibited stability upon thermal, light, storage, and gastrointestinal digestion exposure.	[158]
Fucoxanthin	NPs	Alginate, casein, and chitosan	NPs improved the release of fucoxanthin under simulated gastrointestinal digestion conditions. The membrane permeability of fucoxanthin was enhanced. NPs loaded with fucoxanthin exhibited enhanced plasma levels after oral administration.	[159]
	Nanocomplexes	Whey protein	Nanocomplexes protected fucoxanthin against UV-B radiation, heat, and pH. Enhanced ROS accumulation, caused mitochondrial damage, and regulated apoptosis.	[156]
Lycopene	Nanofibers	Gelatin	Improved water solubility. Enhanced antioxidant activity during 14-day storage.	[160]
	NPs	Alginate phosphatidylcholine	Improved lycopene bioavailability upon cellular uptake by Caco-2 cells. Increased efficiency, stability, and dispersion of lycopene.	[161]
	Lipid-core nanocapsules	Tween 80, span 60, poly-ɛ-caprolactone, and coconut oil	Decreased the viability of MCF-7 cells after 24 and 72 h of exposure. Inhibited the activation of NF-κB and reduced ROS production in microglial (HMC3) cells.	[162]
Lutein	NPs	Stevioside	Entrapment into NPs improved the bioavailability of lutein. Lutein-loaded NPs enter cells by clathrin-mediated endocytosis.	[163]
	Nanoemulsion	Linoleic acid, oleic acid, sodium taurocholate, and mono-oleoyl glycerol	Enhanced the aqueous solubility of lutein. Improved the tissue distribution pattern of lutein in liver and eyes of mice models.	[164]
Crocin	Nanocapsules	Lecithin and chitosan	The sustained release of crocin was promoted. Encapsulation of crocin protective its integrity under in vitro digestion conditions.	[165]
	Nanoemulsion	Chitosan and alginate	Nanoemulsion exhibited high stability under stimulated gastric conditions (pH 2) and executed sustained release of crocin.	[166]

Abbreviations: H_2_O_2_, hydrogen peroxide; DPPH, 2,2-diphenyl-1-picrylhydrazyl; NPs, nanoparticles; UV-B, ultraviolet B; ROS, reactive oxygen species; NF-κB; nuclear factor kappa B.

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
