# Peer review of "Recent Advances in the Therapeutic Potential of Carotenoids in Preventing and Managing Metabolic Disorders"

_plants, 2024, doi:10.3390/plants13121584_

Round 1

Reviewer 1 Report

Comments and Suggestions for Authors

REVIEW OF THE ARTICLE BY ANA E. ORTEGA-REGULES ET AL. ENTITLED “EXPLORING THE THERAPEUTIC POTENTIAL OF CAROTENOIDS IN PREVENTING AND MANAGING METABOLIC DISORDERS: A COMPREHENSIVE REVIEW”

The review summarises the recent background on carotenods, their sources, methods of extraction and analysis. Special attention is paid to their encapsulation and the applications for the treatment of metabolic syndrome. All references are new (2021 and later). Review is actual and it is in the scope of the journal. Very difficult (but mostly correct language) is a drawback. A significant drawback is a distraction from the main point of the text and a high volume of non-related information about the encapsulation. I fill, the review can be published after a revision. There are some scientific mistakes. Please, see the oments below.

L. 33. Typing error.

l. 47. “plants, microorganisms, and animals” - the classification is overlapping, because some microalgae are related to plants and to microorganisms. What is about fungi (macromycetes)? I would rephrase it to “higher plants, algae, bacteria and fungi”. Carotenoids in plants are rather an exception. Its content in some animals is low and they cannot synthesise carotenoids de novo (with one exception). 

l. 50-51. What is the difference between vegetables and foods, fruits and foods, etc?

l. 62.  to the broad class?

l. 66-69. The classification is incomplete. For example, where are the places for crocin, echinenone and canthaxanthin. 

l. 105. Google Scholar is not a database, but a search system.

l. 132. ROS and free radicals are not the same. Singlet oxygen, hydrogen peroxide and nitrous acid are not free radicals. Revise.

l. 177. What is Z-γ-carotene? all-Z-γ-carotene?

l. 173-198. A very important method of CO2 supercritical extraction must be discussed (see e.g. 10.1016/j.tifs.2020.07.016, 10.1016/j.supflu.2017.09.028 or your reference [106]).

l. 188-198. If you mention H. lacustris, you also have to mention other algal sources of carotenoids, since algae is the main source of natural pigments (10.3390/md21020108).

l. 191. Why? As a rule, natural wild-type cultures are used as its source, especially on an industrial scale.

l. 191. If you want to mention astaxanthin production on genetically modified organisms, you can mention transfected animal cells (10.3390/bioengineering10091073).

Use actual names of organisms instead of their obsolete synonyms:

Haematococcus pluvialis - Haematococcus lacustris

l. 209. “column chromatography protocols” - what do you mean? HPLC protocols are also standard.

l. 213. “astaxanthin, keto-carotenoids, and 3-hydroxyechinenone” - revise. astaxanthin and 3-hydroxyechinenone are ketocarotenoids (not keto-carotenoids). What do you mean?

l. 214. As far as I know, Balaustium muroum does not contain astaxanthin.

l. 215. I clearly checked the reference [78], but there was no information about any genetic modifications.

Figure 2. The figure should be revised. Some structures are presented more than once. For some carotenoids optical stereoisomers are shown and for some non-stereoisomeric formulas are. Why? What do you mean when you show specific stereoisomers? Please, comment. Reference to this figure should be given earlier, at the first mention of their structure.

l. 221-232. You can also discuss recent results in carotenoid analysis by FLIM (10.1007/s00709-024-01956-9).

l. 252. Red paprika should not be italicised.

Figure 3. Describe separate panels in the legend in detail.

Section 4. In my opinion, the subsection is far from the main topic of the review. It distracts the reader from the main topic (from the title). It should be shortened. I suggest focusing on the encapsulated forms for MetS treatment. Tables 2-3 also should be shortened and/or combined. The term MetS is used only twice in the section!

l. 331, 372, 410, 427, 467, 509. Add Reference to the figure.

l. 379. Lectins are also proteins.

Comments on the Quality of English Language

In general, English is correct, but difficult. There are many very long sentences. I detected several grammatical errors (see below). Please, check punctuation and articles through the text.

-l. 71. production comprehend - production include?

-l. 77. apoptosis in cancer cells.

-l. 186. these methods

l. 212. The same.

l. 291. from 1-1000 m -> from 1 to 1000 m

l. 305. in the fact that

Author Response

Please see the attachment. Responses are in red.

Reviewer 2 Report

Comments and Suggestions for Authors

Ortega-Regules et al. aim to overview the current scientific knowledge (within last 5 years) on the use of carotenoids towards metabolic syndromes through a systematic overview of metabolic syndrome, followed by summarizing sources, extraction, characterization and biological activity of carotenoids, as well as the need for encapsulating carotenoids and the application of specific microparticles and nanoparticles such as astaxanthin, fucoxanthin, lycopene, lutein, crocin and b-carotene in improving stability, bioavailability, and biological activity are all discussion. This is an interesting review article, which is timely, systematically approached, and well-written. However, the following issues need to be elaborately addressed for a possible publication in ‘Plants’ journal.

1.    The title should be modified as “Recent advances on the therapeutic potential of carotenoids in preventing and managing metabolic disorders”.

2.    Two more keywords should be included; ‘stability’ and ‘bioavailability’.

3.    A bibliometric analysis on the indicated search keywords mentioned in lines 104-106 should be done using ‘Web of Science’ and provide a plot of an increasing number of published papers over the past 5 to 10 years.

4.    Section 3.1 – some more discussion emphasizing the importance of green extraction methods using green solvents by comparing their safety and yield with conventional extraction methods.

5.    Section 3.1 – there is no discussion on UPLC or UHPLC methods for determination of carotenoids and the importance of mass (MS) or tandem mass (MS/MS) detection methods. A comparative approach on HPLC, UHPLC and UPLC methods, as well as detection methods PDA, DAD, MS and MS/MS methods and column types such as C18 and C30.

6.    In sections 4.1 to 4.6, some figures dealing with extraction methods, quantitation methods, stability, bioavailability, and biological activity reported within the last 5 years should be reproduced after obtaining copyright permission. The authors can modify the presentation of reported figures according to their discussion. In this way, the quality of this review article will be greatly enhanced to earn more citations for both the journal and authors.

Comments on the Quality of English Language

Minor editing of English language required

Reviewer 3 Report

Comments and Suggestions for Authors

General comments

·       English language is acceptable and readable so it does not require further improvement.

·       This study presents a literature review of the uses of carotenoids for different diseases and the techniques for improving their physico-chemical properties in detail.

Minor remarks

·       Please, use the already defined abbreviations in the manuscript.

·       Greek symbols should be depicted in italics.

·       All minor remarks are depicted in the manuscript.

Major remarks

·       I recommend better presenting the need for a literature review of carotenoids and their application. So, the aim of this study should be better written and justify this manuscript.

·       Also, the following study DOI: 10.3390/biom11020225 reported using olive oil for beta carotene isolation from orange peels and described their encapsulation in the alginate beads. You can use it for the presentation of the procedure for carotenoid extraction and their stabilization in the literature review.

·       It is not clear why the authors described some of the carotenoids separately.

Comments on the Quality of English Language

The English language is acceptable and readable.

Round 2

Reviewer 1 Report

Comments and Suggestions for Authors

The authors have signifficantly improved the text. There are just minor comments related to the new figures.

Figure 1. I would remove 2024 year, becayse it has not finished.

Figure 2-6. It is too confusing to have subpanels with the same letters.

l. 138. Please, understand, hydrogen peroxyde and hydroxyl radical are also ROS. Please, correct

Reviewer 2 Report

Comments and Suggestions for Authors

The authors have satisfactorily addressed all the comments raised by reviewers and substantially improved the overall quality of the article. Therefore, I recommend accepting this article for publication in Plants.

Comments on the Quality of English Language

Minor editing of English language required

Author Response

Dear reviewer, thank you for your critical review of our work. We strongly thank your comments, which enabled us to improve the quality of the submitted manuscript.